# The feasibility of respondent-driven sampling with people who use drugs in rural Western Cape, South Africa: A qualitative study

Tara Carney [1,2,3*], Kim Johnson[1], Christina Meade[4], Nandi Niemand[5], Jennifer Rooney[6] Sarah Weber[7], Tersius Lambrechts[1], Noluthando Mpisane[1], Charles Robert Horsburgh[8], Danie Theron[9], Robin Warren[5], Sarah Thomson[7], Victoria Overbeck[7], Bronwyn Myers[1,2,10‡], Karen Jacobson[7‡]

**1** Mental Health, Alcohol, Tobacco and Other Drug Research Unit, South African Medical Research Council, Cape Town, South Africa, **2** Department of Psychiatry and Mental Health, University of Cape Town, Cape Town, South Africa, **3** Department of Psychology, University of Johannesburg, Johannesburg, South Africa, **4** Department of Translational Neuroscience, Wake Forest University School of Medicine, Winston-Salem, North Carolina, United States of America, **5** South African Medical Research Council Centre for Tuberculosis Research, Division of Molecular Biology and Human Genetics, Faculty of Medicine and Health Sciences, Stellenbosch University, Cape Town, South Africa, **6** Department of Environmental Health, Harvard University School of Public Health, Boston, Massachusetts, United States of America, **7** Section of Infectious Diseases, Department of Medicine, Boston Medical Center and Boston University School of Medicine, Boston, Massachusetts, United States of America, **8** Department of Epidemiology, Boston University School of Public Health, Boston, Massachusetts, United States of America, **9** Brewelskloof Hospital, Worcester, South Africa, **10** Faculty of Health Sciences, Curtain enAble Institute, Curtain University, Perth, Australia

‡ Shared senior-authorship
* tara.carney@mrc.ac.za

## Abstract

The Western Cape is South Africa's epicentre for tuberculosis (TB) and smoked drug use such as methamphetamine and methaqualone (Mandrax). Despite this, there are limited studies on people who smoke drugs (PWSD) with TB disease in South Africa, partly due to recruitment challenges. Respondent-driven sampling (RDS) is a network-based sampling method used to recruit such key populations. The aim of this qualitative study is to explore the appropriateness and feasibility of RDS as a method for recruiting PWSD for a planned study on TB transmission in this setting. We conducted ten focus group discussions (n = 84) with men and women from Worcester, a rural town in the Western Cape, who self-reported current methamphetamine and/or methaqualone use. Participants were recruited through an existing TB study or community-based outreach. Discussion topics included use of illicit drugs within social networks, feasibility of using RDS methods for recruiting PWSD, and logistical recommendations for the use of RDS and planned study participation. Data were analyzed using thematic analysis. Results indicate drug use by participants across large social networks which is favorable for RDS methods. The key themes were: 1) drug-use social network characteristics including demographic and geographic differences; 2) perspectives of PWSD on RDS methods; 3) potential challenges to proposed RDS

**Data availability statement:** The data collection tool has been included a supplement (interview guide). Since raw qualitative data in the form of transcripts is generally not provided, and our participants are from a key population who provided sensitive information to us as part of this study, and we did not indicate that we would be sharing these in our ethics submission or consent form during dissemination, so would like to discuss an alternative waythis data is not available. Our protocol submitted to the ethics committee in South Africa (Stellenbosch University is the committee on record) also only states that the South African and US investigators will review the transcripts. Please contact Ms Melody Shana, the contact person at Stellenbosch University's Health Research Ethics Committee (HREC1) at melodys@sun.ac.za for further data requests. We have also looked at other recently published qualitative research included in the journal which only includeand it seems only interview or focus group guides have been included as supplementary material.

**Funding:** The funder of this study is the United States National Institute of Diseases (NIAID) (1R01A1147316) (awarded to KJ). The funder had no role in study design, data collection and analysis, decision to publish or preparation of the manuscript.

**Competing interests:** The authors have declared that no competing interests exist.

recruitment and participation in a larger research study for PWSD, and 4) participant recommendations to enhance the uptake of RDS and study participation by PWSD. RDS seems to be a feasible method to recruit PWSD and improve the possibility of reaching a diverse sample of PWSD, with clear recommendations from participants regarding how to recruit participants for larger research studies. The current study indicates that conducting formative, qualitative research can assist researchers with RDS study design and planning for additional study activities.

**Trial Registration:** ClinicalTrials.gov NCT041515602

## Introduction

Globally, tuberculosis (TB) is a prevalent infectious disease with high mortality rates [1], and South Africa is a country with one of the highest burdens of disease [2]. Substance use is associated with an increased risk of TB acquisition and transmission, disease, and treatment failure [3] due in part to poor medication adherence. This is a concern as South Africa has high levels of substance use. The most recent national household survey (conducted in 2017) [4] estimated the prevalence of any drug use in the last three months at 8.6% of the adult population. This is largely driven by cannabis use, followed by methamphetamine (tik) or methaqualone (Mandrax) use, which are often smoked together in the Western Cape. The most recent World Drug Report indicates that methamphetamine use in South Africa is the highest on the African continent [5]. Furthermore, treatment data indicates that an estimated 13.3% of the adult population meet diagnostic criteria for a lifetime substance use disorder, with rates increasing up to 20.6% in the Western Cape [6].

While people who smoke drugs (PWSD) have been identified as a key population for TB detection and control [7], they are often not reached for TB screening due to community and health worker stigma towards PWSD [8,9]. PWSD may experience stigma on numerous levels [10], including in healthcare settings. Stigma is especially prevalent in rural communities with smaller populations and limited treatment options, where people with TB [11] and PWSD may be "known" to health services and the community and have less anonymity [12]. Stigma towards PWSD who are at risk for TB also impacts on their willingness to participate in and engage in health research [13], leading to significant under-representation of this key population group in research which contributes to gaps in our understanding of TB transmission risk among PWSD.

One way to enhance the representation of PWSD in TB research is through peer recruitment methods like Respondent-Driven Sampling (RDS). Peer recruitment may help overcome stigma-related barriers to recruiting this vulnerable group of people for health-related research, which may lead to improved engagement in healthcare as part of these research activities. Peer recruitment has been identified as a feasible way to address substance use and reducing infectious disease risk behaviors in low- and middle-income countries (LMICs) [14].

RDS is a widely used sampling method for collecting representative data from hard-to-reach but socially networked populations [15–17]. For RDS to be successful, "seeds" (original participants) are carefully selected and provided with primary incentives to recruit peers from their social networks [16] through providing them with a predetermined number of coupons to participate in the research (median number = 3 but study dependent) [18]. In turn, if successfully recruited these peers are given a primary incentive to complete study activities and then given the same number of coupons to recruit others from their social network, with the provision of additional 'incentives' for each peer that they have referred who is successful recruited into the study. This process continues until recruitment 'chains' are achieved. Globally, RDS is known for being an efficient and cost effective recruitment method [18] due to its success in reaching the target sample and has been used to recruit high priority populations at risk for HIV and other infectious diseases, including women who engage in sex work, men who have sex with men, and people who use drugs (PWUD) [18]. RDS has also been used to recruit PWSD in urban communities in the Western Cape [19,20].

We aim to conduct a cross-sectional, observational study using RDS recruitment methods to assess TB transmission through illicit drug use networks, focusing on methamphetamine and methaqualone use, in a high TB burden setting and identify mechanisms which underlying accelerate TB transmission [21]. However, the feasibility of using RDS to recruit PWSD from rural communities, particularly in LMICs with high burden of disease, is less understood. In these settings, stigma [11], challenges around social and economic development, racial and other class differences, accessibility, and HIV and TB health literacy may affect study participation. It is therefore key to conduct research to better understand social networks among PWSD in rural communities and ascertain whether it is possible to meet the requirements of RDS as a recruitment method for PWSD. These requirements include ensuring that the population of interest is adequately socially networked and includes relationships of different strengths and types to ensure social diffusion, and that drug use behaviors are spread to different social networks and neighbourhoods [17]. In addition, there should be bridges between separate cliques or sub-networks, to avoid recruitment bottlenecks where different recruitment chains converge on different prevalence estimates or fail to converge at all, which can compromise RDS assumptions for the generation of population parameters [22].

Given these concerns, we aimed to explore the perceptions of PWSD from a rural community within the Western Cape, South Africa on the feasibility and appropriateness of RDS for a study that collects biobehavioral data and will enable the examination of TB transmission within social networks of PWSD in the community mentioned above [21].

This study occurred in the main town of the Breede Valley municipality in the Western Cape Province, South Africa, a prominent wine producing region [23]. The town has an estimated population size of 97 000, with approximately 59% of the population being of mixed race ancestry ('Coloured' according to official population groups), 25% 'Black African' and 14% 'White' [24]. Racial segregation in this community is likely, probably exacerbated by language diversity as most inhabitants speak Afrikaans but there is a growing isiXhosa-speaking population. Worcester is economically disadvantaged, mainly based on the legacy of apartheid, and has a high burden of TB and smoked drug use [7]. In addition, there are high levels of gang violence due to rival gang membership. These factors may hinder RDS, particularly as a recent study found that fear of gang violence impacted on participants' access to healthcare [24]. These contextual influences on health research participation are important to identify and consider when planning for RDS studies. Formative research can ascertain whether peer recruitment of PWSD is possible in this community, and whether the social networks of PWSD meet the criteria for RDS, therefore improving our understanding of the feasibility of RDS-based study in similar rural settings.

## Methods

### Study design

This qualitative study involved ten focus group discussions (FGDs) with PWSD [21]. Eight groups (four conducted in Afrikaans and four conducted in isiXhosa) were initially conducted in 2020 (start date: 04/02/2020), and two in 2022 (end

date: 03/08/2022) to explore recruiting isiXhosa-speaking seed participants for the main RDS study. These groups were stratified by language (Afrikaans and isiXhosa) and self-identified gender. The study is presented according to consolidated criteria for reporting qualitative research (COREQ) guidance [25].

## Participant recruitment and characteristics

Participants were recruited from a large rural town (Worcester) in the Western Cape province. Recruitment occurred through convenience sampling with maximum variation. Participants were recruited through the following ways: 1) an existing TB study which also assessed drug use, 2) referral by friends involved in the study or 3) community-based outreach by research assistants familiar with the community who identified and approached potential participants in areas where smoked drug use was common, and obtained their verbal consent before screening them for study eligibility. Individuals were eligible for study participation if they reported being older than 15 years of age, living in the target community and self-reported methamphetamine or methaqualone use in the past 90 days. If eligible, potential participants were given a date, time and venue to attend the relevant FGD. Of the 99 individuals eligible for study participation, 84 (94.4%) were available and took part in FGDs.

## Focus group procedures

FGDs were held in private, easily accessible venues within the community. Before the FGD started, written informed consent was obtained in the participants' choice of Afrikaans or isiXhosa, the languages predominantly spoken in this community. The size of focus groups ranged between four and ten participants. Experienced qualitative researchers (Masters-level training) facilitated the group discussions while other study staff assisted with taking notes. To avoid any bias in data collection methods, facilitators were not involved in participant recruitment and had no prior relationships with participants. FGDs lasted an average of 80 minutes and were audio-recorded with participant consent. FGDs were transcribed verbatim by study staff with transcription experience and who spoke Afrikaans and isiXhosa. These transcripts were then translated into English using professional transcribers who specialize in language services for data analysis. A semi-structured guide was used to facilitate the FGDs (S1 Text). This guide was developed with input from community members, namely a community advisory board and key research staff in South Africa and then revised with feedback from US investigators, to ensure questions were appropriate for the population and addressed the gaps in working with PWSD at-risk of TB. It comprised open-ended questions and probes about participants' composition and size of social networks, substance use within this network, perspectives on RDS methods (including peer recruitment, biomedical specimen collection and fingerprints to avoid repeat enrollment), barriers to research participation among PWSD, and acceptability and feasibility of RDS and study activities in the larger planned study (see [21] for detailed procedures on proposed RDS study activities). Participants received grocery vouchers [to the value of ZAR150] to thank them for their time and participation. Participants were also offered a referral to substance use treatment.

## Data analysis

We used thematic analysis to analyze the data [26]. The first two FGDs were coded independently using NVivo software by the first and third author (both experienced in qualitative data analysis) before meeting to review codes generated and resolve discrepancies. The remaining transcripts were double coded with coders meeting with the first author (TC) to iteratively develop the codebook. All coding discrepancies were resolved by discussion between coders, with the first author available to break any coding ties. Interrater reliability was measured throughout the coding process, with a final Kappa score of 0.89. Coding continued iteratively as new themes emerged from the transcripts, with the codebook modified as new concepts, which did not fit into the existing coding scheme, were identified. Following several rounds of coding, codes

were collated (after discussion with TC) and developed into initial themes. After completing the first stages of analysis, two of the authors presented key preliminary findings to members of the community at an organized meeting to check accuracy. While no new findings emerged, they agreed that the presented findings were representative of social networks of drug use and RDS study participation barriers and facilitators in the larger community.

## Ethical considerations

Stellenbosch University [N19/10/128], and Boston Medical Centre [H-38910] granted ethics approval, and the South African Medical Research Council granted approval for the University of Stellenbosch to be the Ethics Committee of Record (S2 Text).

## Results

FGD participants' demographic characteristics are provided below (Table 1).

Four themes were generated. These are presented below and illustrated with quotes from the FGDs. The first is *characteristics of social networks of PWSD*, which describes how social networks of PWSD cut across demographic and geographic differences, which is important for RDS feasibility. The next two themes describe *participants' perspectives of RDS methods* and *potential challenges to the use of RDS and participation in larger research study activities*, namely barriers to recruitment of PWSD through RDS, with a specific sub-theme on *violence and safety*. The final theme addresses participants' *recommendations for enhancing the uptake of RDS and study participation*.

**Table 1. Demographic Profile of Focus Group Participants.**

|  | Overall (n = 84) |
|---|---|
| **Age** | |
| Median [Min, Max] | 28 [25, 32] |
| **Self-Reported Gender** | |
| Male | 44 (52.4%) |
| Female | 39 (46.4%) |
| Missing | 1 (1.2%) |
| **Self-Reported Ethnicity** | |
| Coloured[a] | 28 (33.3%) |
| Black African | 55 (65.5%) |
| Other | 1 (1.2%) |
| **Language** | |
| Afrikaans | 31 (36.9%) |
| isiXhosa | 41 (48.8%) |
| English | 12 (14.3%) |
| **Recruitment and Referral Process** | |
| Existing research study | 25 (28.8%) |
| Friend | 8 (9.5%) |
| Community outreach | 50 (59.5%) |
| Missing | 1 (1.2%) |

[a] often defined as of mixed race ancestry, descended from Khoisan people who originally inhabited the western parts of South Africa, as well Asian and African slaves brought to the Cape, from Europe, and from other African countries.

## Social network characteristics of PWSD

Participants discussed their social networks in which they used drugs. The size of these networks varied from two to three individuals with whom they had well-established relationships to larger social networks of up to 160 people (including close and more casual relationships). For most participants, close relationships formed the foundation of their drug use networks. These participants described using drugs with a group of people with whom they had close ties, such as "a circle of friends" or "crew", family members, and sexual partners.

> "Some girls smoke with their boyfriends. You see your boyfriend wandering and every time you find him in this area. And then the problem now is he will try to get rid of you and sit with other smokers and other smokers are girls. So you end up joining them because you want to be in this place with your man." [Female participant, isiXhosa FGD]

In addition to these established relationships, participants described casual relationships that formed part of their drug-use social network on occasion. These included "acquaintances" or "smoking friends" and "friends until the drugs are finished" who were often simply members of the community who also smoked drugs. These expedient relationships were based on the need to pool money to afford drugs and sometimes extended participants' social networks to include a more diverse group of individuals in terms of age, gender, and ethnic backgrounds.

> "If we can now buy a lolly [methamphetamine glass pipe], [and] we have R50 [$2.6] for a little bag, if it's in a half hours' time, or five minutes time, or in two days' time, we are going to smoke when the opportunity arises. Like I say again, drugs do not ask colour, age, who. If a 15-year-old comes to me now, me who is 42, I am going to tell him, make the pipe, burn there. You want to smoke." [Male participant, Afrikaans FGD]

These expedient relationships also introduced neighbourhood diversity into participant drug-use networks. Participants discussed how they travelled to other geographic locations and communities to buy drugs when there was a shortage in their own community:

> "I know you stay there in [Neighbourhood X], but I am in [Neighbourhood Y]. If here now is not [drugs], then I go to you there. But your price is now just a little bit cheaper. But I want it, so I'm going to do the effort. If I hear now that here in Worcester there are no buttons (methaqualone), only in [next community], then I will go there..." [Male participant, Afrikaans FGD]

However, for the most part, this diversity in their drug-use social network did not extend to type or mode of substance use. Only a few participants even mentioned the inclusion of people who injected drugs [PWID] in their drug-use social network, and one reason provided for not smoking drugs with PWID was due to the because the magnitude and duration of the drugs' effect if injected:

> "For example, she has a R50 bag and I have a R50 bag and it's the same quality. And say I am on the injection and she's on the smoke, I'm going to feel higher than her. Or it goes through quicker." [Female participant, Afrikaans FGD]

## Participants' perspectives of RDS methods

When it came to peer recruitment, participants mentioned that this would be possible with the provision that detailed information on the study was provided to potential seed participants to ensure that the recruitment procedure was transparent. Some participants suggested that this could be included on the recruitment coupon:

*"I would say it is alright, some of us we are not good in explaining, at least it's gonna help us to cut it short when we are explaining. You must be able to talk, explain, for the people to understand what you are talking about. You must be able to influence people."* [Male participant, isiXhosa FGD]

According to most participants, PWSD would be eager to participate in a larger RDS research study if there was a space for participants to speak about their substance use without judgement and if they could obtain support for their drug use and referral to treatment if needed during study participation. Not only was this about having an available physical space but also having an understanding, supportive person to speak to about their drug use:

*"People that use drugs are very shy. No one must know about their things…Let's say many are withdrawn for one or the other thing. You must not know a lot about my things. But the things we are talking to you now, that we don't actually share, that's why they won't talk about it a lot. But because I am comfortable to talk about it, I share it, because it is for a study, I can maybe save the next guy."* [Male participant, Afrikaans FGD]

## Potential challenges to the use of RDS and participation in larger research study activities

Participants described numerous potential barriers when recruiting PWSD through RDS as well as participation in the larger RDS study, which includes multiple levels of stigma, fear of legal repercussions of drug use, and hesitation to undergo biological tests.

Several participants expressed fear that accepting a recruitment coupon would identify them as PWSD and that they would be "targeted" by police. Participants thought other PWSD would be concerned about the potential legal ramifications of providing biometric data (such as the use of fingerprints to avoid repeat enrollments) as part of the study recruitment procedures and identity verification, particularly for those with a history of other illegal behavior. These participants had the perception that the police or justice system would be able to access participants' study information:

*"But you must think like this now, now the police car stands [waits] there …. They know what goes on in this venue. Now tomorrow I'm going to feel like a target, that woman saw me that day I'm going to a study, she's going to want to shake me out [try and obtain information about drug use]….* [Female participant, Afrikaans FGD]

*"People might be concerned if you take their fingerprints. They might have a criminal record, or they might be running away from police. So those might be their concerns."* [Male participant, isiXhosa FGD]

In terms of participation in the larger study activities, participants cited stigma as a reason for often not disclosing drug use to people outside of their social networks of PWSD and being hesitant to join research studies and even attend healthcare services. Participants mentioned experiencing "gossip" and name calling by community members, including healthcare providers, which resulted in them feeling "shame" and "embarrassment". This seemed to be exacerbated by their friends' and families' reactions once they were made aware of the participants' drug use:

*"Even our parents, they don't understand, they just shout at you for smoking…tik [methamphetamine]. They don't sit you down and ask 'why are you smoking ', and you also reply by shouting too. Our parents are unable to sit us down and ask what we like and we don't like …. I will continue with my doings [drug use] because it is known now. While arriving at home they will say 'here arrives ipharaphara [the parasite]'. Everyone knows me as such."* [Male participant, isiXhosa FGD]

In addition, participants thought biological testing for drug use would serve as a barrier to study participation. Apart from the fear of needles for blood draw purposes, participants thought PWSD would feel shame and embarrassment about

positive drug tests results and its confirmation of use to study staff, and fear that their results would not remain confidential but be discovered by the broader community.

*"They will never attend, they know that they will be exposed that they are using drugs, they don't want to be told about their status….the urine will indicate that they are using drugs."* [Female participant, isiXhosa FGD]

**Violence and safety.** Another potential challenge to study enrollment, even once recruitment coupons have been accepted, was widespread gang activity and violence (often viewed as drug-related) in the community. When discussing potential study site locations, most participants described an inability to enter certain neighbourhoods due to "gang territory" and "rivalry" or even "war". This was especially the case for male participants who might be suspected of gang membership even if they had no gang affiliations:

*"He will not be able to visit his friends on [that] community as he will be robbed or stabbed. There is also gangsterism; you just get involved in such things. And you won't be able to go and buy tik [methamphetamine] in [neighbourhood X] while they are selling it here [in this community] because you will get hurt on your way from the selling place, it is better to buy where you stay."* [Male participant, isiXhosa FGD]

Therefore, a suggestion from several participants was to find a central, neutral location for study activities which would be associated with less gang activity and would therefore be more accessible to study participants from a variety of neighborhoods. A more central location was also recommended to promote the anonymity of study participants, as they would not be "seen" by people within or close to their neighborhood.

*"All these areas are not safe. It is good that one is in the middle or in the centre. And not move to other places, because those places are not safe."* [Female, isiXhosa FGD]

*"And it's far from us….which is better, because at least you won't live [there]… you leave for longer and not care what gossipers say behind your back."* [Female participant, isiXhosa FGD]

However, a study site in a central location was viewed by other participants as challenging due to opportunistic crime when crossing neighborhood boundaries to access the site, such as being robbed or beat, even by other study participants:

*"I am thinking that sometimes guys also attend the study, so if people walk [to the site], guys might think of robbing them, take their phones, laptops. So, it would also be a disadvantage because they might see an opportunity to rob… There are people who smoke, who have been in jail for years who are not afraid or even scared to go back to jail. So, it would be easy for those people to rob."* [Female participant, isiXhosa FGD]

### Recommendations for enhancing the uptake of RDS and study participation

Participants shared their perceptions on ways to address the challenges to reach seed participants and their social network of PWSD through RDS. This included ensuring participants' information is kept confidential, the study site being a safe space for PWSD to discuss their drug use but also in terms of community violence, transparency around study procedures and suggestions on logistical issues. First, to address concerns around legal consequences of information collected as part of study requirements, participants strongly recommended that study staff clearly explain the RDS recruitment method and other study procedures, specifically the collection of fingerprints during study enrollment to avoid repeat enrollment:

*"Maybe I might refuse to do finger printing because I would want to come again tomorrow. Like I would change my hair style wear my afro hair, wig the next day and put make up so that you don't realise it's me, get the second voucher."* [Female participant, isiXhosa FGD]

To overcome fears around drug use disclosure, participants highlighted the importance of confidentiality around study participation and biological test results:

*"Because here at the clinic to come for my blood [tests] people would say... You see she was here... I'm sitting and people are talking about me and they don't understand that I came for what I came for. Now here [at the clinic] you will be told about [person's name] that you have seen here."* [Male participant, isiXhosa FGD]

Some participants also mentioned that explaining the benefits of study-specific biological tests, such as for HIV and TB detection, to participants would provide them with insights into their health status and reduce their fears. Participants spoke specifically about PWSD who were "serious about taking care of your health":

*"Look, there with [another study] they took blood many times and it helps a person, because if there's maybe an illness that you have, then you can now find out you have that illness. Do you understand? But if you must now go to a doctor to get that blood tests looked at and all that things, it's going to cost a lot of money".* [Male participant, Afrikaans FGD]

Finally, participants had several suggestions about study logistics. This included the best time to reach and work with PWSD when trying to recruit seed participants. Late mornings and early afternoons were viewed by most as acceptable times, and the consensus was that weekdays would be most effective, apart from Fridays or "scratch" day, when individuals "itched" to use drugs, and with drug use continuing through the weekend:

*"We only count on Fridays, here the weekend starts now. The weekends start on Fridays. Saturdays and Fridays and Sundays."* [Male participants, Afrikaans FGD]

In addition, participants thought that the incentive provided to participants when enrolling or recruiting additional peers to participate in the study would be very attractive to PWSD from this community. Grocery vouchers were suggested to assist future participants with food purchases, and most participants felt that the value of these vouchers offered should reflect the value of time spent during study visits and the number of proposed specimens collected:

*"Okay people will come because we have basic needs that need money, it may help me to buy such things like [sanitary] pads, maybe you don't have food, or anything you might need, that R150 is a lot at least you can buy something to eat or helps you to travel or buy cosmetics."* [Female participant, isiXhosa FGD]

## Discussion

This study highlights the importance of conducting formative, qualitative research to inform the use of RDS for recruiting PWSD from a rural area in the Western Cape for a larger study focused on understanding TB transmission in this population. For the most part, participants described diverse and differentiated social networks of PWSD that would facilitate RDS, with these social networks sometimes based on expedient relationships that were formed to address drug use needs. Participants expressed that peer recruitment through RDS methods would be possible in this setting. However, this study also identified potential challenges when using RDS as a recruitment strategy for participation in a larger research study. Challenges included concerns about drug use disclosure and subsequent experiences of stigma, community violence, and the collection of biometric data and biological samples as part of identity and drug use status verification, which

are critical components for RDS. Together these concerns may impact the willingness of PWSD to participate in RDS for research. However, participants also recommended ways of overcoming these challenges to facilitate peer recruitment and research-related TB and other health assessments of PWSD.

The study findings confirm that the recruitment of peers through social networks may offer an opportunity to reach a diverse population of PWSD and highlight the feasibility of conducting RDS in this population. According to Johnson et al [16], for RDS to be feasible, the target population's social networks of PWSD need to be large and diverse in terms of strengths and types. The findings of the current study indicated that participants who smoked drugs indeed had large, diverse social networks in terms of closeness of relationships and sociodemographic factors. The findings of this study are in agreement with previous studies conducted in LMICs, which have looked at the size [19], composition and strengths of ties within social networks of PWSD [27]. Previous RDS studies in LMICs have found that this kind of mixing occurs between groups [28] typically divided by sociodemographic characteristics such as gender, age, and race, as well as geographic location [29] between neighborhoods within their broader community [30].

However, in the current qualitative study, there was limited evidence of diversity in the mode of drug use, as the inclusion of PWID in certain social networks of PWSD was only mentioned by a minority of participants. This is representative of the fact that smoking is the most common mode of drug use in the Western Cape, with injection drug use being less prevalent based on treatment data [31]. These kinds of relationships between subgroups (PWSD and PWID) may be particularly important in the larger planned RDS study, which investigates the transmission of infectious diseases, including TB (and HIV co-morbidity). PWID have been identified as a key population at risk for TB [32,33], especially in LMICs [34,35]; therefore it will be important to consider how to include this population, perhaps in future studies, despite that this being a much smaller subgroup of people who use drugs in this setting.

The study findings extend our understanding of barriers to research participation for PWSD [17] by highlighting barriers that may be specific to studies that use RDS methods to recruit participants. Although the results indicated that peer recruitment seemed feasible, one of the main issues around participating in health research mentioned was stigma against PWSD within their broader communities, interpersonal relationships, and healthcare settings. Participants described how stigma affected PWSD's ability to communicate openly about their drug use and disclose this, which may affect their willingness to participate in studies like the proposed larger study. Stigma has been consistently documented as a barrier to study participation using RDS recruitment methods with key populations such as PWID [34,35], men who have sex with men [36,37], transgender women [37] and women who engage in sex work [38,39]. To address the anticipated fear of stigma, FGD participants suggested providing a safe space for participants, defined as a centrally located study site where they would not be judged by staff or the community. This is consistent with previous studies in this setting, where the characteristics of study staff have been important for overcoming barriers to research participation [40]. Recent research has found that professionalism combined with empathy is critical to engaging with PWSD [41,42] in non-stigmatizing ways, and that peers can be instrumental in engaging with PWSD living with TB and HIV [42].

The current findings also underscore the importance of being aware of other challenges to the recruitment and participation of PWSD in studies using RDS recruitment methods in rural settings. Some of these included concerns around the provision of personal information and biological specimens, and physical safety of study participation in a community where gang violence and criminal activity are rife across different neighbourhoods. Gang rivalry and territory issues are associated with safety concerns [24]. Participants themselves provided solutions to overcoming some of these barriers, including logistical recommendations for study participation that will be important to consider when planning larger studies that use RDS methods. Finally, the FGDs highlighted the heterogeneity of PWSD in this rural community and their competing priorities. The latter is particularly relevant for informing participant reimbursement rates given the economic disadvantage in this setting where even basic needs were reportedly difficult to meet. Studies using RDS methods will need to consider how to appropriately compensate participants for their research involvement (in line with national guidelines [43]) while ensuring that the value of these vouchers does not amount to coercion.

Findings should be considered of the study's limitations. First, not all FGDs were conducted before the launch of the RDS study. While this allowed us to explore actual challenges to recruitment and how these could be addressed, participants in the second round of FGDs already knew about the study activities in detail which could have affected their feedback in that they would comment specifically on their actual experiences in the study, as opposed to providing more general feedback. Another limitation was that the original FGD transcripts were conducted in local languages to enable participants to communicate freely, and then transcribed and translated into English before coding. It is possible that some themes or vernacular terms specific to these languages could have been slightly changed during translation to English. However, since high quality translation practices were used, this risk was mitigated as much as possible. Another study limitation is that the study sample constituted a small number of PWSD who resided in one rural community in the Western Cape, and therefore, may not have been representative of PWSD in other types of communities in other provinces of South Africa. However, the authors feel that describing the study context and results in rich detail will allow readers to assess the transferability of findings to their own setting.

There were also methodological strengths to this study. FGDs included participants who self-identified both as male or female to provide balanced gender responses as well as participants who spoke both Afrikaans and isiXhosa to ensure representation of various neighbourhoods. Another strength of the study was that the recommendations put forward by study participants were utilized to inform the research procedures of larger study that used RDS recruitment methods. For example, the study site location was identified after considering issues of safety and centrality, staff members underwent training on stigma prevention and the discussion sensitive topics when working with key populations such as gang activity and substance use, and recruitment materials were designed to ensure transparency. Recruitment days and times, and grocery incentives, were determined based on feedback from these FGDs.

In conclusion, although it is not viewed as strictly necessary to conduct formative work to inform RDS procedures, this study highlights the benefits of conducting qualitative research when using RDS to reach a key population in a new setting. We strongly recommend that other research teams planning to use RDS as a recruitment strategy conduct formative work to inform their study procedures, particularly when conducting research within a new setting or population. The inputs from PWSD confirmed that RDS was largely an acceptable and feasible way of recruiting PWSD in this setting, and highlighted opportunities to refine proposed research procedures to enhance feasibility and the importance of considering barriers to the use of RDS that may exist in the social context before starting recruitment. This formative work has helped to ensure that study procedures planned for an approved RDS study [21] were acceptable to potentially eligible PWSD.

## Supporting information

**S1 Text. Focus Group Discussion Guide.**
(PDF)

**S2 Text. Ethics Approval Documents.**
(DOCX)

**S3 Text. Inclusivity in Global Research.**
(DOCX)

## Acknowledgments

The authors thank our study team, and study participants without whom our work would not be possible. We would also like to thank the community spaces that allowed us to conduct this research in settings where our participants felt secure and safe.

See S3 Text for information on the Inclusivity in Global Research.

## Author contributions

**Conceptualization:** Tara Carney, Jennifer Rooney, Charles Robert Horsburgh, Robin Warren, Karen Jacobson.

**Formal analysis:** Kim Johnson, Noluthando Mpisane, Victoria Overbeck, Bronwyn Myers.

**Funding acquisition:** Charles Robert Horsburgh, Karen Jacobson.

**Investigation:** Nandi Niemand, Robin Warren.

**Methodology:** Tara Carney, Christina Meade, Jennifer Rooney, Sarah Thomson, Tersius Lambrechts, Robin Warren.

**Project administration:** Kim Johnson, Nandi Niemand, Sarah Thomson, Tersius Lambrechts, Danie Theron, Sarah Weber, Victoria Overbeck.

**Visualization:** Bronwyn Myers.

**Writing – original draft:** Tara Carney, Noluthando Mpisane, Bronwyn Myers.

**Writing – review & editing:** Tara Carney, Kim Johnson, Christina Meade, Nandi Niemand, Jennifer Rooney, Sarah Thomson, Tersius Lambrechts, Charles Robert Horsburgh, Danie Theron, Robin Warren, Sarah Weber, Victoria Overbeck, Karen Jacobson.

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
