## [Decision Letter · Decision Letter 0]

PGPH-D-24-01788

The feasibility of respondent-driven sampling with people who use drugs in rural Cape Town, South Africa: a qualitative study

Dear Dr. Carney,

Thank you for submitting your manuscript to PLOS Global Public Health. After careful consideration, we feel that it has merit but does not fully meet PLOS Global Public Health’s publication criteria as it currently stands. Therefore, we invite you to submit a revised version of the manuscript that addresses the points raised during the review process.

Both reviewers agree that the paper has met some of the PLoS Global Health requirements for publication but could undergo revision to best present your data and analysis. Reviewers have asked, and I agree, that the paper sharpen its analytical framework and its use of interpretive work to help let your argument come to the fore more clearly. Similarly one reviewer has asked that you better foreground your methods so as to help readers understand the data on which you base your claims. Moreover, reviewer 1 has several very useful suggestions about how the authors might highlight their research objectives and then better orient their methodological choices around these objectives. I think this is excellent advice and encourage the authors to follow it. Similarly, reviewer 1 has suggested that the authors might be more explicit about the research as implementation research or as descriptive research. Both are useful orientations but being a bit more clear about how your tools were developed and what they intend to do will be useful as you revise the paper. Both reviewers have created lists of small queries about the text and a careful engagement with these issues will help strengthen the article's overall argument. Finally, please not that reviewer 1's review is attached to this email as a file.

We look forward to receiving your revised manuscript.

Kind regards,

Andrew James McDowell

Academic Editor

Journal Requirements:

Additional Editor Comments (if provided):

Reviewers' comments:

Reviewer's Responses to Questions

**Comments to the Author**

1. Does this manuscript meet PLOS Global Public Health’s publication criteria?

Reviewer #1: Partly

Reviewer #2: Yes

2. Has the statistical analysis been performed appropriately and rigorously?

Reviewer #1: N/A

Reviewer #2: N/A

3. Have the authors made all data underlying the findings in their manuscript fully available (please refer to the Data Availability Statement at the start of the manuscript PDF file)?

Reviewer #1: Yes

Reviewer #2: No

4. Is the manuscript presented in an intelligible fashion and written in standard English?

Reviewer #1: Yes

Reviewer #2: No

Reviewer #1: The paper requires a major revision to meet the standards for publication. The objectives are not clear, thus it is unknown if the data presented supports the objectives and the conclusions. A great deal of methodological information is also missing

Reviewer #2: This is an interesting paper, and I congratulate the authors. Aside of its relevance with regard to testing a methodology, the paper gives great insight of what may be viewed as an emic view of this group as they live, create and mold relationships, and meet their drug needs while navigating the penal system and the public moral terrain, and their own identities as citizens and human beings. The paper shows a rich side to this groups lifeworld, desires, vulnerabilities, and resiliencies.

I am not sure how this framing could be woven into this paper, so that these rich lives of a group pursuing agency are foregrounded, and how effort to bring RDS does appear to offset (although at the same time it may disrupt positively) these lives. I would wish to see some interpretive texts in the findings and also another couple in the discussion. The issues such as refusal of fingerprints so that one may come in disguise and get another voucher demonstrate compellingly this group’s need for resources and invite a careful consideration of where the line might lie between reimbursing / incentivizing on one hand, and exploiting vulnerabilities on the other.

Below I make a few suggestions and highlight some typos.

Abstract

1. Use respondent-driven or respondent driven consistently

2. Ten FGD (n=84) – how ‘n’ is used here is not clear; may I suggest specifying in this way or similar ‘total number of participants’ = 84.

3. You could specify the kind of place or level of organization Worcester is in SA (city, province, suburb?

4. Specify in the abstract how you actually recruited these participants, and their demographics too.

5. In the results: I would suggest specifying the number and areas of themes first (before starting to describe actual results as has been done in the opening line),

6. I would like to suggest not just listing the themes, but actually summarizing key findings therein, to help your conclusions and findings to align.

7. Specify, as you do in the main paper, that the recommendations to enhance the uptake of RDS by PWSD’, came from participants, not from the researchers.

Text

1. Line 75: Please give citation for household survey

2. Check grammar – “as a potentially feasible methods”

3. Grammar Ln 99: “and it known for its efficiency and cost effectiveness [“

4. Ln 100: Not clear what “and potentially for participants” is referring to

5. Ln 101 “PWSD urban communities of PWSD”

6. Ln 105. You refer to ‘increased stigma’ when talking about rural areas; did you have citations that point to this?

7. Ln 109: please check “97,00”

8. Since you describe the demographic composition of the study, could you provide some breakdown of subgroups living there

9. Line 122.You mention one of the requirements of RDs as ‘ensuring that the population of interest (for example PWSD) is adequately socially networked and allows relationships of different strengths and types to ensure social diffusion” Could you please elaborate regarding social diffusion, and the context in which you refer to the specific paper (the paper does not seem to explicitly mention social diffusion).

10. Ethical issues do not seem a little out of place under design.

11. Ln 147 – when you say historically disadvantaged, could you please specify for an international audience.

12. In the paragraph on recruitment (Ln 145-154), if you could specify who was screened for eligibility, was it all potential participants (including those identified through friends from the study, and those identified by study staff by entering the community).

13. The table is not included in the draft paper

14. Ln 163 – Please elaborate on the significance of ‘Facilitators were not involved in participant recruitment and had no prior relationships with participants”

15. Ln 167” Please elaborate how community input into the FGD guide was obtained

16. Ln 184 = “Two of the authors presented key findings to members of the community to ensure accuracy shortly after the majority of FGDs were held in 2020.” Was this after much of the analysis had been done, or after only partial analysis?

17. Could you please review the ordering of your first 2-3 paras of findings. You mention in para 1 causal relationships and larger networks of up to 160 people. However, a little further below (ln 208-212) you seem to imply you have not referred prior to casual groups. Could you try to find a way to tie this part together.

18. The earlier part of the findings appears to be a form of mapping, and it might help to actually present these forms or relationships in a figure of some sort. I would understand, though, if it turned out to be difficult conceiving the figure. Please take this suggestion at your discretion.

19. Check spelling in ln 227

20. Lines 231-234 are hard to follow, maybe due to the use of negatives. Please consider rephrasing.

21. Ln 264 Plea review the grammar, police car stand’. If ‘stand’ is verbatim, I would suggest using a more correct or common term in parenthesis. Also, what does ‘going to a study’ and ‘shake me out’ mean?

22. Is there any plausible explanation for the PWUD not wanting to the evidence of their use in the bloods.

23. Please review quote in ln 326 -329 to make clearer

24. Check grammar in paragraph starring 331

25. Among limitations you mention that discussions were conducted in the local language resulting in potential to lose themes during translation. As a limitation, this is not clear to me; often local language is encouraged, and the researchers would need to ensure staff are competent or translation is available.

26. I also learnt towards the end of the paper, that the RDS seemed to already have been implemented at this time of submitting this paper. Is this correct? Reference to this seems to be in passing. It may help to make clearer much earlier that the RDS study was subsequently carried out, and that the full reporting of findings will be done separately. This may help to contextualized some of very rich and telling findings presented in this analysis.

**Do you want your identity to be public for this peer review?** For information about this choice, including consent withdrawal, please see our Privacy Policy

Reviewer #1: No

Reviewer #2: **Yes: ** Jeremiah Chikovore

---

## [Decision Letter · Decision Letter 1]

PGPH-D-24-01788R1

The feasibility of respondent-driven sampling with people who use drugs in rural Cape Town, South Africa: a qualitative study

Dear Dr. Carney,

Thank you for submitting your manuscript to PLOS Global Public Health. After careful consideration, we feel that it has merit but does not fully meet PLOS Global Public Health’s publication criteria as it currently stands. Therefore, we invite you to submit a revised version of the manuscript that addresses the points raised during the review process.

Please carefully review the revisions suggested by the reviewer, which are attached to this letter. 

The reviewer has raised a number of concerns that require attention, including suggestions regarding the themes and overall aims of the paper. The reviewer also requests additional clarification of various points throughout your manuscript. 

Could you please revise the manuscript and provide a point-by-point response to these comments?

We look forward to receiving your revised manuscript.

Kind regards,

Sarah Jose, Ph.D.

Staff Editor

Additional Editor Comments (if provided):

Reviewers' comments:

Reviewer's Responses to Questions

**Comments to the Author**

Reviewer #1: (No Response)

publication criteria?

Reviewer #1: Partly

3. Has the statistical analysis been performed appropriately and rigorously?

Reviewer #1: N/A

4. Have the authors made all data underlying the findings in their manuscript fully available (please refer to the Data Availability Statement at the start of the manuscript PDF file)?

Reviewer #1: Yes

5. Is the manuscript presented in an intelligible fashion and written in standard English?

Reviewer #1: Yes

Reviewer #1: Congratulations to the authors for their work on this important topic, including the edits to the previous draft. The manuscript is improved following your revisions, however I believe more work is required before the manuscript can be published. The most notable issue to improve the coherence between the stated objectives, the titles of the themes in the results section, and the data. Copy editing is also required to check references, remove extra words, and improve sentence clarity. Other issues are outlined in my attached review.

**Do you want your identity to be public for this peer review?** For information about this choice, including consent withdrawal, please see our Privacy Policy

Reviewer #1: No

---

## [Decision Letter · Decision Letter 2]

PGPH-D-24-01788R2

The feasibility of respondent-driven sampling with people who use drugs in rural Cape Town, South Africa: a qualitative study

Dear Dr. Carney,

Thank you for submitting your manuscript to PLOS Global Public Health. After careful consideration, we feel that it has merit but does not fully meet PLOS Global Public Health’s publication criteria as it currently stands. Therefore, we invite you to submit a revised version of the manuscript that addresses the points raised during the review process.

We look forward to receiving your revised manuscript.

Kind regards,

Jianhong Zhou

Staff Editor

Journal Requirements:

Additional Editor Comments (if provided):

Reviewers' comments:

Reviewer's Responses to Questions

**Comments to the Author**

Reviewer #1: (No Response)

publication criteria?

Reviewer #1: Yes

3. Has the statistical analysis been performed appropriately and rigorously?

Reviewer #1: N/A

4. Have the authors made all data underlying the findings in their manuscript fully available (please refer to the Data Availability Statement at the start of the manuscript PDF file)?

Reviewer #1: Yes

5. Is the manuscript presented in an intelligible fashion and written in standard English?

Reviewer #1: Yes

Reviewer #1: (No Response)

**Do you want your identity to be public for this peer review?** For information about this choice, including consent withdrawal, please see our Privacy Policy

Reviewer #1: No

---

## [Decision Letter · Decision Letter 3]

The feasibility of respondent-driven sampling with people who use drugs in rural Cape Town, South Africa: a qualitative study

PGPH-D-24-01788R3

Dear Dr. Carney,

We are pleased to inform you that your manuscript 'The feasibility of respondent-driven sampling with people who use drugs in rural Cape Town, South Africa: a qualitative study' has been provisionally accepted for publication in PLOS Global Public Health.

Best regards,

Julia Robinson

Executive Editor

Reviewer Comments (if any, and for reference):

Reviewer's Responses to Questions

**Comments to the Author**

Reviewer #1: All comments have been addressed

publication criteria?

Reviewer #1: Yes

3. Has the statistical analysis been performed appropriately and rigorously?

Reviewer #1: N/A

4. Have the authors made all data underlying the findings in their manuscript fully available (please refer to the Data Availability Statement at the start of the manuscript PDF file)?

Reviewer #1: Yes

5. Is the manuscript presented in an intelligible fashion and written in standard English?

Reviewer #1: Yes

Reviewer #1: as per attached file.

**Do you want your identity to be public for this peer review?** For information about this choice, including consent withdrawal, please see our Privacy Policy

Reviewer #1: No
